# RenderNet: A deep convolutional network for differentiable rendering from 3D shapes

**Thu Nguyen-Phuoc**
University of Bath
T.Nguyen.Phuoc@bath.ac.uk

**Chuan Li**
Lambda Labs
c@lambdalabs.com

**Stephen Balaban**
Lambda Labs
s@lambdalabs.com

**Yong-Liang Yang**
University of Bath
Y.Yang@cs.bath.ac.uk

## Abstract

Traditional computer graphics rendering pipelines are designed for procedurally generating 2D images from 3D shapes with high performance. The non-differentiability due to discrete operations (such as visibility computation) makes it hard to explicitly correlate rendering parameters and the resulting image, posing a significant challenge for inverse rendering tasks. Recent work on differentiable rendering achieves differentiability either by designing surrogate gradients for non-differentiable operations or via an approximate but differentiable renderer. These methods, however, are still limited when it comes to handling occlusion, and restricted to particular rendering effects. We present RenderNet, a differentiable rendering convolutional network with a novel projection unit that can render 2D images from 3D shapes. Spatial occlusion and shading calculation are automatically encoded in the network. Our experiments show that RenderNet can successfully learn to implement different shaders, and can be used in inverse rendering tasks to estimate shape, pose, lighting and texture from a single image.

## 1 Introduction

Rendering refers to the process of forming a realistic or stylized image from a description of the 3D virtual object (e.g., shape, pose, material, texture), and the illumination condition of the surrounding scene (e.g., light position, distribution, intensity). On the other hand, inverse rendering (graphics) aims at estimating these properties from a single image. The two most popular rendering methods, rasterization-based rendering and ray tracing, are designed to achieve fast performance and realism respectively, but not for inverse graphics. These two methods rely on discrete operations, such as z-buffering and ray-object intersection, to identify point visibility in a rendering scene, which makes these techniques non-differentiable. Although it is possible to treat them as non-differentiable renderers in computer vision tasks [1], inferring parameters, such as shapes or poses, from the rendered images using traditional graphics pipelines is still a challenging task. A differentiable renderer that can correlate the change in a rendered image with the change in rendering parameters therefore will facilitate a range of applications, such as vision-as-inverse-graphics tasks or image-based 3D modelling and editing.

Recent work in differentiable rendering achieves differentiability in various ways. Loper and Black [2] propose an approximate renderer which is differentiable. Kato et al. [3] achieve differentiability by proposing an approximate gradient for the rasterization operation. Recent work on image-based reconstruction uses differentiable projections of 3D objects onto silhouette masks as a surrogate for a rendered image of the objects [4, 5]. Wu et al. [6] and Tulsiani et al. [7] derive differentiable projective

functions from normal, depth, and silhouette maps, but respectively can only handle orthographic projection, or needs multiple input images. These projections can then be used to construct an error signal for the reconstruction process. All of these approaches, however, are restricted to specific rendering styles (rasterization) [2, 3, 8], input geometry types [9, 10], or limiting output formats such as depth or silhouette maps [4, 5, 6, 7, 11, 12]. Moreover, none of these approaches try to solve the problem from the network architecture design point of view. Recent progress in machine learning shows that network architecture plays an important role for improving the performance of many tasks. For example, in classification, ResNet [13] and DenseNet [14] have contributed significant performance gains. In segmentation tasks, U-Net [15] proves that having short-cut connections can greatly improve the detail level of the segmentation masks. In this paper, we therefore focus on designing a neural network architecture suitable for the task of rendering and inverse rendering.

We propose RenderNet, a convolutional neural network (CNN) architecture that can be trained end-to-end for rendering 3D objects, including object visibility computation and pixel color calculation (shading). Our method explores the novel idea of combining the ability of CNNs with inductive biases about the 3D world for geometry-based image synthesis. This is different from recent image-generating CNNs driven by object attributes [16], noise [17], semantic maps [18], or pixel attributes [19], which make very few assumption about the 3D world and the image formation process. Inspired by the literature from computer graphics, we propose the projection unit that incorporates prior knowledge about the 3D world, and how it is rendered, into RenderNet. The projection unit, through learning, is a differentiable approximation of the non-differentiable visibility computation step, making RenderNet an end-to-end system. Unlike non-learnt approaches in previous work, a learnt projection unit uses deep features instead of low-level primitives, making RenderNet generalize well to a variety of input geometries, robust to erroneous or low-resolution input, as well as enabling learning multi-style rendering with the same network architecture. RenderNet is differentiable and can be easily integrated to other neural networks, benefiting various inverse rendering tasks, such as novel-view synthesis, pose prediction, or image-based 3D shape reconstruction, unlike previous image-based inverse rendering work that can recover only part of the full 3D shapes [20, 21].

We choose the voxel presentation of 3D shapes for its regularity and flexibility, and its application in visualizing volumetric data such as medical images. Although voxel grids are traditionally memory inefficient, computers are becoming more powerful, and recent work also addresses this inefficiency using octrees [22, 23], enabling high-resolution voxel grids. In this paper, we focus on voxel data, and leave other data formats such as polygon meshes and unstructured point clouds as possible future extensions. We demonstrate that RenderNet can generate renderings of high quality, even from low-resolution and noisy voxel grids. This is a significant advantage compared to mesh renderers, including more recent work in differentiable rendering, which do not handle erroneous inputs well.

By framing the rendering process as a feed-forward CNN, RenderNet has the ability to learn to express different shaders with the same network architecture. We demonstrate a number of rendering styles ranging from simple shaders such as Phong shading [24], suggestive contour shading [25], to more complex shaders such as a composite of contour shading and cartoon shading [26] or ambient occlusion [27], some of which are time-consuming and computationally expensive. RenderNet also has the potential to be combined with neural style transfer to improve the synthesized results, or other complex shaders that are hard to define explicitly.

In summary, the proposed RenderNet can benefit both rendering and inverse rendering: RenderNet can learn to generate images with different appearance, and can also be used for vision-as-inverse-graphics tasks. Our main contributions are threefold.

- A novel convolutional neural network architecture that learns to render in different styles from a 3D voxel grid input. To our knowledge, we are the first to propose a neural renderer for 3D shapes with the projection unit that enables both rendering and inverse rendering.
- We show that RenderNet generalizes well to objects of unseen category and more complex scene geometry. RenderNet can also produce textured images from textured voxel grids, where the input textures can be RGB colors or deep features computed from semantic inputs.
- We show that our model can be integrated into other modules for applications, such as texturing or image-based reconstruction.

## 2 Related work

Our work is related to three categories of learning-based works: image-based rendering, geometry-based rendering and image-based shape reconstruction. In this section, we review some landmark methods that are closely related to our work. In particular, we focus on neural-network-based methods.

**Image-based rendering** There is a rich literature of CNN-based rendering by learning from images. Dosovitskiy et al. [16] create 2D images from low-dimensional vectors and attributes of 3D objects. Cascaded refinement networks [18], and Pix2Pix [28] additionally condition on semantic maps or sketches as inputs. Using a model that is more deeply grounded in computer graphics, DeepShading [19] learns to create images with high fidelity and complex visual effects from per-pixel attributes. DC-IGN [29] learns disentangled representation of images with respect to transformations, such as out-of-plane rotations and lighting variations, and thus is able to edit images with respect to these factors. Relevant works on novel 3D view synthesis [30] leverage category-specific shape priors and optical flow to deal with occlusion/disocclusion. While these methods yield impressive results, we argue that geometry-based methods, which make stronger assumptions about the 3D world and how it produces 2D images, will be able to perform better in certain tasks, such as out-of-plane rotation, image relighting, and shape texturing. This also coincides with Rematas et al. [31], Yang et al. [32] and Su et al. [33] who use strong 3D priors to assist the novel-view synthesis task.

**Geometry-based rendering** Despite the rich literature in rendering in computer graphics, there is a lot less work using differentiable rendering techniques. OpenDR [2] has been a popular framework for differentiable rendering. However, being a more general method, it is more strenuous to be integrated into other neural networks and machine learning frameworks. Kato et al. [3] approximate the gradient of the rasterization operation to make the rendering differentiable. However, this method is limited to rasterization-based rendering, making it difficult to represent more complex effects that are usually achieved by ray tracing such as global illumination, reflection, or refraction.

**Image-based 3D shape reconstruction** Reconstructing 3D shape from 2D image can be treated as estimating the posterior of the 3D shape conditioned on the 2D information. The prior of the shape could be a simple smoothness prior or a prior learned from 3D shape datasets. The likelihood term, on the other hand, requires estimating the distribution of 2D images given the 3D shape. Recent work has been using 2D silhouette maps of the images [4, 5]. While this proves effective, silhouette images contain little information about the shape. Hence a large number of images or views of the object is required for the reconstruction task. For normal maps and depth maps of the shape, Wu et al. [6] derive differentiable projective functions assuming orthographic projection. Similarly, Tulsiani et al. [7] propose a differentiable formulation that enables computing gradients of the 3D shape given multiple observations of depth, normal or pixel color maps from arbitrary views. In our work, we propose RenderNet as a powerful model for the likelihood term. To reconstruct 3D shapes from 2D images, we do MAP estimation using our trained rendering network as the likelihood function, in addition to a shape prior that is learned from a 3D shape dataset. We show that we can recover not only the pose and shape, but also lighting and texture from a single image.

## 3 Model

The traditional computer graphics pipeline renders images from the viewpoint of a virtual pin-hole camera using a common perspective projection. The viewing direction is assumed to be along the negative z-axis in the camera coordinate system. Therefore, the 3D content defined in the world coordinate system needs to be transformed into the camera coordinate system before being rendered. The two currently popular rendering methods, rasterization-based rendering and ray tracing, procedurally compute the color of each pixel in the image with two major steps: testing visibility in the scene, and computing shaded color value under an illumination model.

RenderNet jointly learns both steps of the rendering process from training data, which can be generated using either rasterization or ray tracing. Inspired by the traditional rendering pipeline, we also adopt the world-space-to-camera-space coordinate transformation strategy, and assume that the camera is axis-aligned and looks along the negative z-axis of the volumetric grid that discretizes the input shape. Instead of having the network learn operations which are differentiable and easy to implement, such as rigid-body coordinate transformation or the interaction of light with surface

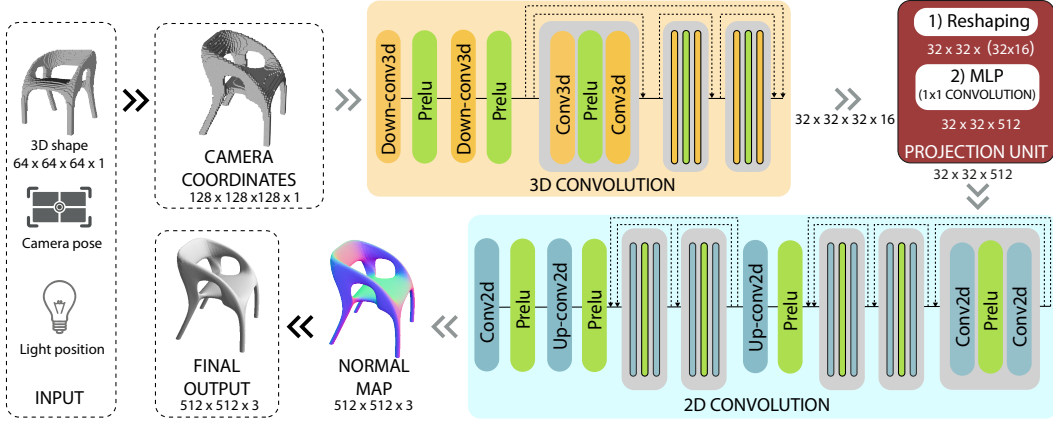

Figure 1: **Network architecture.** See Section 2 in the supplementary document for details.

normals (e.g. assuming a Phong illumination model [24]), we provide most of them explicitly to the network. This allows RenderNet to focus its capacity on more complex aspects of the rendering task, such as recognizing visibility and producing shaded color.

RenderNet receives a voxel grid as input, and applies a rigid-body transformation to convert from the world coordinate system to the camera coordinate system. The tranformed input, after being trilinearly sampled, is then fed to a CNN with a projection unit to produce a rendered 2D image. RenderNet consists of 3D convolutions, a projection unit that computes visibility of objects in the scene and projects them onto 2D feature maps, followed by 2D convolutions to compute shading.

We train RenderNet using a pixel-space loss between the target image and the output. Optionally, the network can produce normal maps of the 3D input which can be combined with light sources to illuminate the scene. While the projection unit can easily incorporate orthographic projections, the 3D convolutions can morph the scene and allows for perspective camera views. In future versions of RenderNet, perspective transformation may also be explicitly incorporated into the network.

## 3.1 Rotation and resampling

The transformed input via rigid body motion ensures that the camera is always in the same canonical pose relative to the voxel grid being rendered. The transformation is parameterized by the rotation around the y-axis and z-axis, which corresponds to the azimuth and elevation, and a distance $R$ that determines the scaling factor, i.e., how close the object is to the camera. We embedded the input voxel grid into a larger grid to make sure the object is not cut off after rotation. The total transformation therefore includes scaling, rotation, translation, and trilinear resampling.

## 3.2 Projection unit

The input of RenderNet is a voxel grid $V$ of dimension $H_V \times W_V \times D_V \times C_V$ (corresponding to height, width, depth, and channel), and the output is an image $I$ of dimension $H_I \times W_I \times C_I$ (corresponding to height, width and channel). To bridge the disparity between the 3D input and 2D output, we devise a novel projection unit. The design of this unit is straightforward: it consists of a reshaping layer, and a multilayer perceptron (MLP). Max pooling is often used to flatten the 3D input across the depth dimension [4, 5], but this can only create the silhouette map of the 3D shape. The projection unit, on the other hand, learns not only to perform projection, but also to determine visibility of different parts of the 3D input along the depth dimension after projection.

For the reshaping step of the unit, we collapse the depth dimension with the feature maps to map the incoming 4D tensor to a 3D squeezed tensor $V'$ with dimension $W \times H \times (D \cdot C)$. This is immediately followed by an MLP, which is capable of learning more complex structure within the local receptive field than a conventional linear filter [13]. We apply the MLP on each $(D \cdot C)$ vector, which we implement using a $1 \times 1$ convolution in this project. The reshaping step allows each unit of the MLP to access the features across different channels and the depth dimension of the input, enabling the

network to learn the projection operation and visibility computation along the depth axis. Given the squeezed 3D tensor $V'$ with $(D \cdot C)$ channels, the projection unit produces a 3D tensor with $K$ channels as follows:

$$I_{i,j,k} = f \left( \sum_{dc} w_{k,dc} \cdot V'_{i,j,dc} + b_k \right) \qquad (1)$$

where $i$, $j$ are pixel coordinates, $k$ is the image channel, $dc$ is the squeezed depth channel, where $d$ and $c$ are the depth and channel dimension of the original 4D tensor respectively, and $f$ is some non-linear function (parametric ReLU in our experiments).

### 3.3   Extending RenderNet

We can combine RenderNet with other networks to handle more rendering parameters and perform more complex tasks such as shadow rendering or texture mapping. We model a conditional renderer $p(I \mid V, h)$ where $h$ can be extra rendering parameter such as lights, or spatially-varying parameters such as texture.

Here we demonstrate the extensibility of RenderNet using the example of the Phong illumination model [24]. The per-pixel shaded color for the images is calculated by $S = \max(0, \vec{l} \cdot \vec{n} + a)$, where $\vec{l}$ is the unit light direction vector, $\vec{n}$ is the normal vector, whose components are encoded by the RGB channels of the normal map, and $a$ is an ambient constant. Shading $S$ and albedo map $A$ are further combined to create the final image $I$ based on $I = A \odot S$ [34]. This is illustrated in Section 4.1, where we combine the albedo map and normal map rendered by the combination of a texture-mapping network and RenderNet to render shaded images of faces.

## 4   Experiments

To explore the generality of RenderNet, we test our method on both computer graphics and vision tasks. First, we experiment with different rendering tasks with varying degree of complexity, including challenging cases such as texture mapping and surface relighting. Second, we experiment with vision applications such as image-based pose and shape reconstruction.

**Datasets**   We use the chair dataset from ShapeNet Core [35]. Apart from being one of the categories with the largest number of data points (6778 objects), the chair category also has large intra-class variation. We convert the ShapeNet Dataset to $64 \times 64 \times 64$ voxel grids using volumetric convolution [36]. We randomly sampled 120 views of each object to render training images at $512 \times 512$ resolution. The elevation and azimuth are uniformly sampled between $[10, 170]$ degrees and $[0, 359]$ degrees, respectively. Camera radius are set at 3 to 6.3 units from the origin, with the object's axis-aligned bounding box normalized to 1 unit length. For the texture mapping tasks, we generate 100,000 faces from the Basel Face Dataset [37], and render them with different azimuths between $[220, 320]$ degrees and elevations between $[70, 110]$ degrees. We use Blender3D to generate the Ambient Occlusion (AO) dataset, and VTK for the other datasets. For the contour dataset, we implemented the pixel-based suggestive contour [25] algorithm in VTK.

**Training**   We adopt the patch training strategy to speed up the training process in our model. We train the network using random spatially cropped samples (along the width and height dimensions) from the training voxel grids, while keeping the depth and channel dimensions intact. We only use the full-sized voxel grid input during inference. The patch size starts as small as 1/8 of the full-sized grid, and progressively increases towards 1/2 of the full-sized grid at the end of the training.

We train RenderNet using a pixel-space regression loss. We use mean squared error loss for colored images, and binary cross entropy for grayscale images. We use the Adam optimizer [38], with a learning rate of 0.00001.

Code, data and trained models will be available at: `https://github.com/thunguyenphuoc/RenderNet`.

### 4.1   Learning to render and apply texture

Figure 2 shows that RenderNet is able to learn different types of shaders, including Phong shading, contour line shading, complex multi-pass shading (cartoon shading), and a ray-tracing effect (Ambient

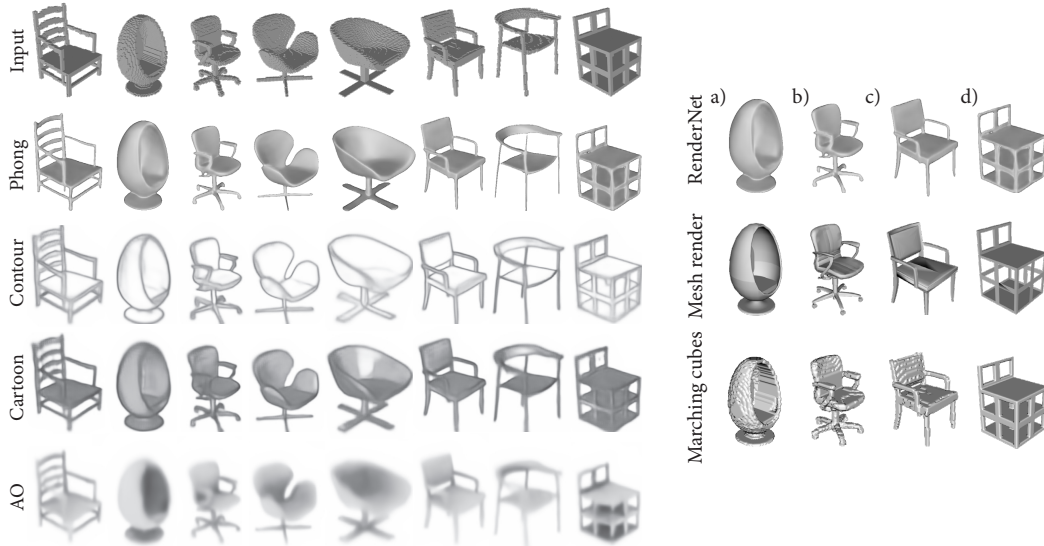

Figure 2: **Left:** Different types of shaders generated by RenderNet (intput at the top). **Right:** Comparing Phong shading between RenderNet, a standard OpenGL mesh renderer, and a standard Marching Cubes algorithm. RenderNet produces competitive results with the OpenGL mesh renderer without suffering from mesh artefacts (notice the seating pad of chair (c) or the leg of chair (d) in Mesh renderer), and does not suffer from low-resolution input like Marching cubes.

Occlusion) with the same network architecture. RenderNet was trained on datasets for each of these shaders, and the figure shows outputs generated for unseen test 3D shapes. We report the PSNR score for each shader in Figure 5.

RenderNet generalizes well to shapes of unseen categories. While it was trained on chairs, it can also render non-man-made objects such as the Stanford Bunny and Monkey (Figure 3). The method also works very well when there are multiple objects in the scene, suggesting the network recognizes the visibility of the objects in the scene.

RenderNet can also handle corrupted or low-resolution volumetric data. For example, Figure 3 shows that the network is able to produce plausible renderings for the Bunny when the input model was artificially corrupted by adding 50% random noise. When the input model is downsampled (here we linearly downsampled the input by 50%), RenderNet can still render a high-resolution image with smooth details. This is advantageous compared to the traditional computer graphics mesh rendering, which requires a clean and high-quality mesh in order to achieve good rendered results.

It is also straightforward to combine RenderNet with other modules for tasks such as mapping and rendering texture (Figure 4). We create a texture-mapping network to map a 1D texture vector representation (these are the PCA coefficients for generating albedo texture using the BaselFace dataset) to a 3D representation of the texture that has the same width, height and depth as the shape input. This output is concatenated along the channel dimension with the input 3D shape before given RenderNet to render the albedo map. This is equivalent to assigning a texture value to the corresponding voxel in the binary shape voxel grid. We also add another output branch of 2D convolutions to RenderNet to render the normal map. The albedo map and the normal map produced by RenderNet are then combined to create shaded renderings of faces as described in Section 3.3. See Section 2.3 in the supplementary document for network architecture details.

## 4.2 Architecture comparison

In this section, we compare RenderNet with two baseline encoder-decoder architectures to render Phong-shaded images. Similar to RenderNet, the networks receive the 3D shape, pose, light position and light intensity as input. In contrast to RenderNet, the 3D shape given to the alternative network is in the canonical pose, and the networks have to learn to transform the 3D input to the given pose. The first network follows the network architecture by Dosovitskiy et al. [16], which consists of a

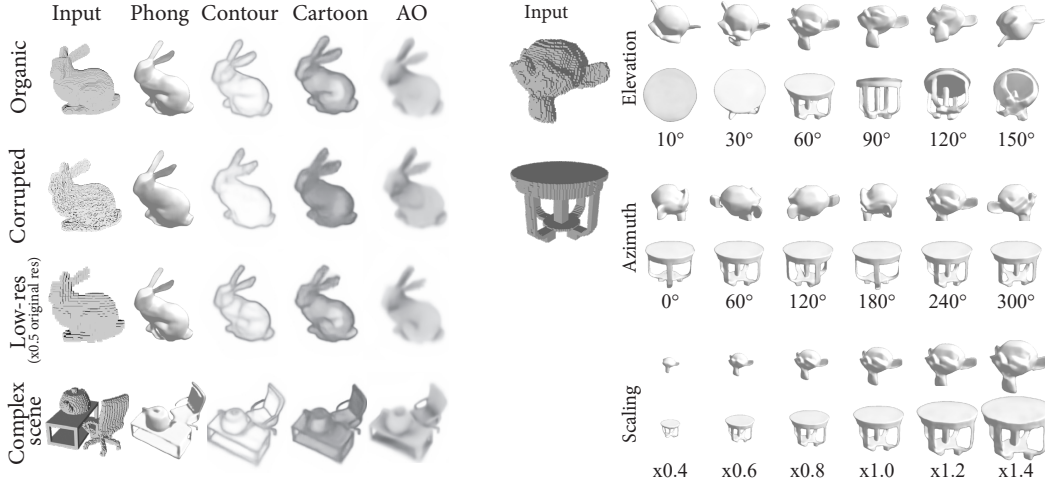

Figure 3: **Generalization.** Even with input from unseen categories or of low quality, RenderNet can still produce good results in different styles (left) and from different views (right).

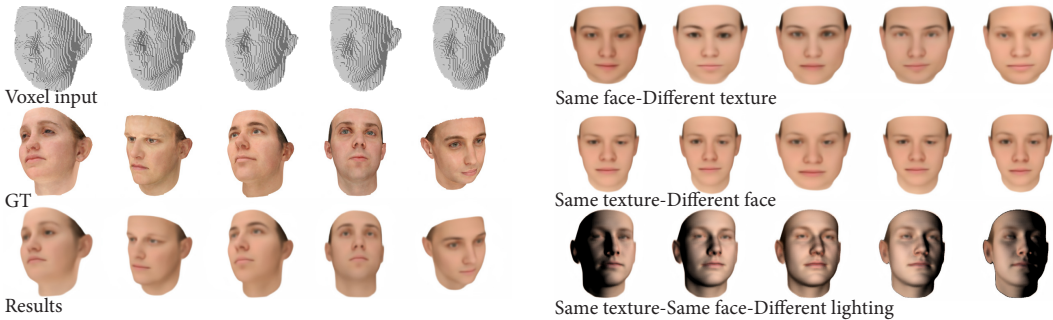

Figure 4: Rendering texture and manipulating rendering inputs. Best viewed in color.

series of fully-connected layers and up-convolution layers. The second network is similar but has a deeper decoder than the first one by adding residual blocks. For the 3D shape, we use an encoding network to map the input to a latent shape vector (refer to Section 2.2 in the supplementary document for details). We call these two networks EC and EC-Deep, respectively. These networks are trained directly on shaded images with a binary cross-entropy loss, using the chair category from ShapeNet. RenderNet, on the other hand, first renders the normal map, and combines this with the lighting input to create the shaded image using the shading equation in Section 3.3.

As shown in Figure 5, the alternative model (here we show the EC model) fails to produce important details of the objects and achieves lower PSNR score on the Phong-shaded chair dataset. More importantly, this architecture "remembers" the global structure of the objects and fails to generalize to objects of unseen category due to the use of the fully connected layers. In contrast, our model is better for rendering tasks as it generalizes well to different categories of shapes and scenes.

### 4.3 Shape reconstruction from images

Here we demonstrate that RenderNet can be used for single-image reconstruction. It achieves this goal via an iterative optimization that minimizes the following reconstruction loss:

$$\underset{z,\theta,\phi,\eta}{\text{minimize}} \quad \|I - f(z, \theta, \phi, \eta)\|^2 \tag{2}$$

where $I$ is the observed image and $f$ is our pre-trained RenderNet. $z$ is the shape to reconstruct, $\theta$ and $\eta$ are the pose and lighting parameters, and $\phi$ is the texture variable. In essence, this process maximizes the likelihood of observing the image $I$ given the shape $z$.

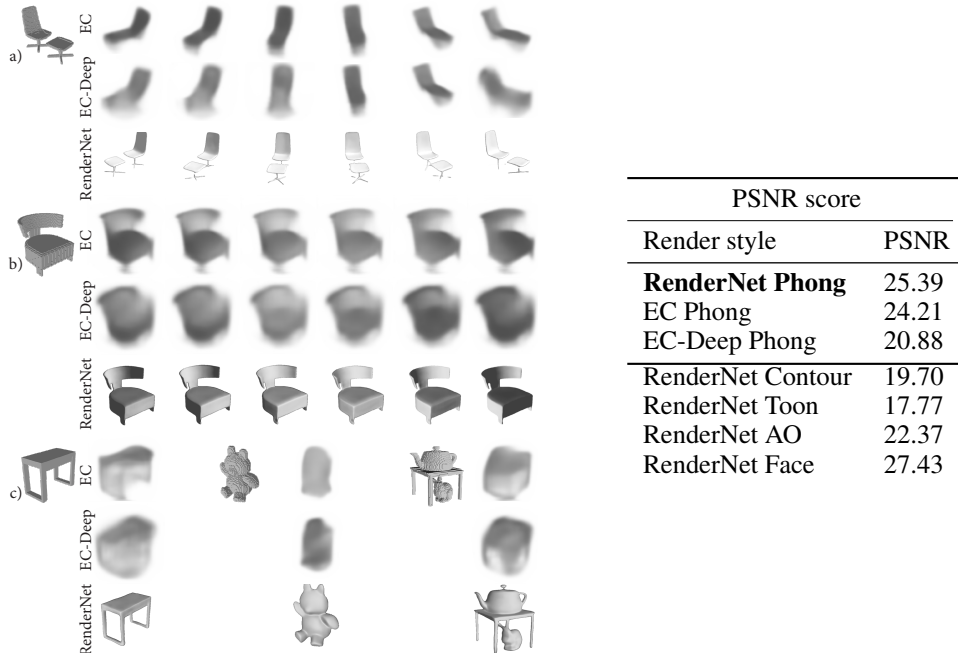

| PSNR score | |
|---|---|
| Render style | PSNR |
| **RenderNet Phong** | 25.39 |
| EC Phong | 24.21 |
| EC-Deep Phong | 20.88 |
| RenderNet Contour | 19.70 |
| RenderNet Toon | 17.77 |
| RenderNet AO | 22.37 |
| RenderNet Face | 27.43 |

Figure 5: **Left:** Architecture comparison in different tasks: **a)** Novel-view synthesis, **b)** Relighting and **c)** Generalization. **Right:** PSNR score of different shaders, including the two alternative architectures.

However, directly minimizing this loss often leads to noisy, unstable results (shown in Figure 2 in the supplementary document). In order to improve the reconstruction, we use a shape prior for regularizing the process – a pre-trained 3D auto-encoder similar to the TL-embedding network [39] with 80000 shapes. Instead of optimizing $z$, we optimize its latent representation $z'$:

$$\underset{z',\theta,\phi',\eta}{\text{minimize}} \quad \|I - f(g(z'),\theta,h(\phi'),\eta)\|^2 \tag{3}$$

where $g$ is the decoder of the 3D auto-encoder. It regularizes the reconstructed shape $g(z')$ by using the prior shape knowledge (weights in the decoder) for shape generation. Similarly, we use the decoder $h$ that was trained with RenderNet for the texture rendering task in Section 4.1 to regularize the texture variable $\phi'$. This corresponds to MAP estimation, where the prior term is the shape decoder and the likelihood term is given by RenderNet. Note that it is straightforward to extend this method to the multi-view reconstruction task by summing over multiple per-image losses with shared shape and appearance.

We compare RenderNet with DC-IGN by Kulkarni et al. [29] in Figure 6. DC-IGN learns to decompose images into a graphics code $Z$, which is a disentangled representation containing a set of latent variables for shape, pose and lighting, allowing them to manipulate these properties to generate novel views or perform image relighting. In contrast to their work, we explicitly reconstruct the 3D geometry, pose, lighting and texture, which greatly improves tasks such as out-of-plane rotation, and allows us to do re-texturing. We also generate results with much higher resolution ($512\times512$) compared to DC-IGN ($150\times150$). Our results show that having an explicit reconstruction not only creates sharper images with higher level of details in the task of novel-view prediction, but also gives us more control in the relighting task such as light color, brightness, or light position (here we manipulate the elevation and azimuth of the light position), and especially, the re-texturing task.

For the face dataset, we report the Intersection-over-Union (IOU) between the ground truth and reconstructed voxel grid of $42.99 \pm 0.64$ for 95% confidence interval. We also perform the same experiment for the chair dataset – refer to Section 1 in the supplementary material for implementation details and additional results.

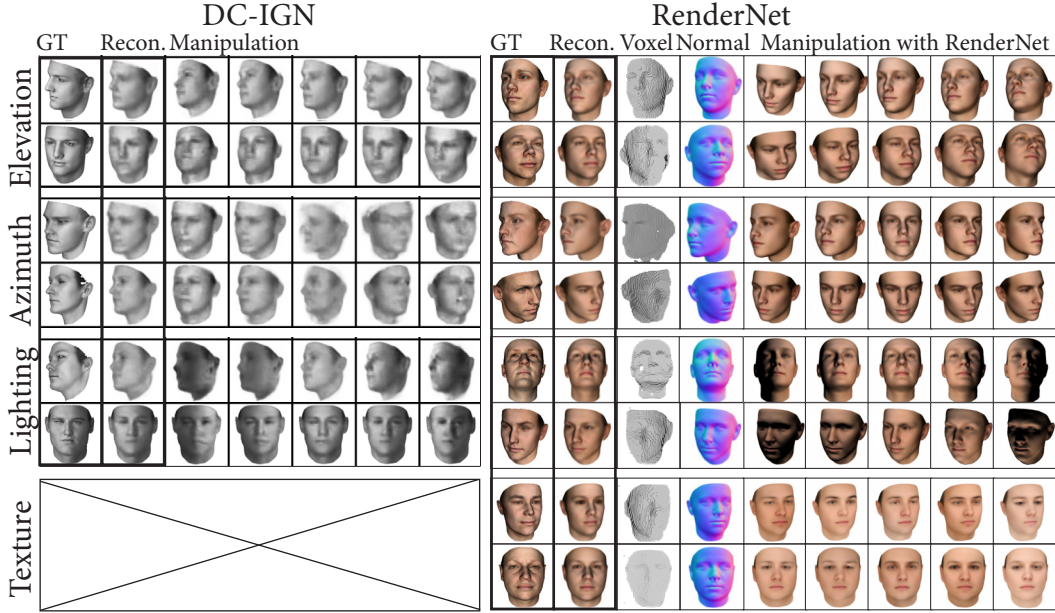

Figure 6: **Image-based reconstruction.** We show both the reconstructed images and normal maps from a single image. The cross indicates a factor not learnt by the network. Note: for the re-texturing task, we only show the albedo to visualize the change in texture more clearly. Best viewed in color.

## 5 Discussion and conclusion

In this paper, we presented RenderNet, a convolutional differentiable rendering network that can be trained end-to-end with a pixel-space regression loss. Despite the simplicity in the design of the network architecture and the projection unit, our experiments demonstrate that RenderNet successfully performs rendering and inverse rendering. Moreover, as shown in Section 4.1, there is the potential to combine different shaders in one network that shares the same 3D convolutions and projection unit, instead of training different networks for different shaders. This opens up room for improvement and exploration, such as extending RenderNet to work with unlabelled data, using other losses like adversarial losses or perceptual losses, or combining RenderNet with other architectures, such as U-Net or a multi-scale architecture where the projection unit is used at different resolutions. Another interesting possibility is to combine RenderNet with a style-transfer loss for stylization of 3D renderings.

The real world is three-dimensional, yet the majority of current image synthesis CNNs, such as GAN [17] or DC-IGN [29], only operates in 2D feature space and makes almost no assumptions about the 3D world. Although these methods yield impressive results, we believe that having a more geometrically grounded approach can greatly improve the performance and the fidelity of the generated images, especially for tasks such as novel-view synthesis, or more fine-grained editing tasks such as texture editing. For example, instead of having a GAN generate images from a noise vector via 2D convolutions, a GAN using RenderNet could first generate a 3D shape, which is then rendered to create the final image. We hope that RenderNet can bring more attention to the computer graphics literature, especially geometry-grounded approaches, to inspire future developments in computer vision.

### Acknowledgments

We thank Christian Richardt for helpful discussions. We thank Lucas Theis for helpful discussions and feedback on the manuscript. This work was supported in part by the European Union's Horizon 2020 research and innovation programme under the Marie Sklodowska-Curie grant agreement No 665992, the UK's EPSRC Centre for Doctoral Training in Digital Entertainment (CDE), EP/L016540/1, and CAMERA, the RCUK Centre for the Analysis of Motion, Entertainment Research and Applications, EP/M023281/1. We also received GPU support from Lambda Labs.

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
