[Supplementary Material]

# *Supplementary material for* RenderNet: A deep convolutional network for differentiable rendering from 3D shapes

In this document, we provide more details for the image-based reconstruction task, and additional results for the chair dataset (Section 1). We also provide more details for different network architectures (Section 2). In Section 3, we comment on the current limitations of RenderNet.

## 1  Image-based reconstruction

### 1.1  Details on the optimization process

We optimize for the loss function:

$$\underset{z',\theta,\phi',\eta}{\text{minimize}} \quad ||I - f(g(z'),\theta,h(\phi'),\eta)||^2 \tag{1}$$

where $I$ is the observed image, $z'$ is a latent vector representation of the 3D shape, $g$ is the decoder of an autoencoder used to learn prior information about shapes (See Section 1.3), $\theta$ is the pose parameter, $\eta$ is the lighting parameter, $\phi'$ is the texture vector and $h$ is the texture decoder. $z'$ is a 200-dimensional vector in this experiment.

For the pose parameter $\theta$, since we only observe faces from the frontal hemisphere, we subdivide the pose space of [0–180] degrees for azimuth, [0–180] degrees for elevation into a grid and use the grid points for initialization. The grid is later further subdivided around current best pose parameters. To avoid local minima, we initialize multiples of $(z'_i, \theta_i, \phi'_i, \eta_i)$ ($i \in \{1, 2, \ldots, 5\}$ in our experiment) and use gradient descent for optimizing all of the variables. We re-initialise the parameters with the current best ones after every 200 steps, and continue with the optimization until convergence, which takes around 1800 steps.

### 1.2  Chair reconstruction from a single image

We optimize for the loss function:

$$\underset{z,\theta}{\text{minimize}} \quad \alpha||I - f(g(z'),\theta)||^2 + \beta(z-\mu)^T \Sigma^{-1}(z-\mu) \tag{2}$$

where $I$ is the observed image, $z'$ is a latent vector representation of the 3D shape, $g$ is the decoder of an autoencoder used to learn prior information about shapes, $\theta$ is the pose parameter, $\mu$ and $\Sigma$ are the mean and covariance of $z'$ estimated from the training set respectively, and $\alpha$ and $\beta$ are the weights of the loss terms (we use $\alpha = 5$, $\beta = 1$). $z'$ is a 250-dimensional vector in this experiment. We also compare with DC-IGN [1], however, we could not download the same dataset [2] that was used for DC-IGN due to broken download links. Therefore, we use the chair category from ShapeNet, which is very similar and greatly overlaps with the dataset used in DC-IGN, as a substitute in this experiment. We use these chair models to create greyscale shaded images used as inputs for the reconstruction task.

We adopt the same optimisation strategy as with the face reconstruction (grid subdivision and gradient descent for the pose and shape vector). The optimisation converges after 2000 steps. The results are shown in Figure 1. Reconstructing chairs is a much more challenging task than reconstructing faces,

Figure 1: Reconstructing chairs from a single image, compared to DC-IGN [1]. The crosses indicate factors not learnt by the network. We were able to recover both the pose and shape of the chairs, which can be used to achieve sharper results in the task of novel-view synthesis, as well as enabling image relighting.

due to the larger search space for the pose parameter ([0–360] for azimuth, [0–180] for elevation), as well as the larger variance in the geometry of different chairs, especially those containing very thin parts, that might not be fully captured by the shape prior. However, for a task as challenging as simultaneous shape and pose estimation, the results show great potential of our method, instead of using a feed-forward network similar to the work of Tulsiani et al. [3]. Further work is needed to improve the speed and performance of this method.

### 1.3   3D shape autoencoder for learning shape prior

We train an antoencoder to learn a prior of 3D shapes. The encoder is a series of 3D convolutions with channels {64, 128, 256, 512}, kernel sizes {5, 5, 2, 2}, and strides {2, 2, 2, 2} respectively. The fully-connected layer in the middle maps the output of the last convolution layer to a 200-dimensional vector. This is followed by a sigmoid activation function and another fully-connected layer that maps the 200-dimensional vector to a $(4 \cdot 4 \cdot 512)$-dimensional vector. This vector is then reshaped to a tensor of size $4 \times 4 \times 512$ before being fed to a series of 3D up-convolutions with channels {256, 128, 64, 32, 1}, kernel sizes {4, 4, 4, 4, 4}, and strides {2, 2, 2, 2, 1}. Here we use ELU activation functions for all layers, apart from the last convolution layer in the encoder and decoder, which uses sigmoid functions.

### 1.4   Reconstruction without prior

To show the importance of using the shape prior $g(z')$, here we compare the reconstruction results between those with the shape prior and those without, i.e., we directly optimize for the shape $z$ without using the prior $g(z')$. Figure 2 shows that the reconstruction without using the shape prior fails to generate good results.

Figure 2: Comparison between the reconstruction results with/without prior.

## 2 Network Architecture

All of our layers use parametric Relu (PReLU) [4], apart from the last layer, which uses a sigmoid function. We also use dropouts with the probability of 0.5 during training after every convolution, except those used in the residual blocks.

Each 3D residual block consists of a $3\times3\times3$ 3D convolution, a PReLU activation function, and another $3\times3\times3$ 3D convolution. The input to the block is then added to the output of the second convolution (shortcut connection). Each 2D residual block is similar to the 3D one, but we replace 3D convolutions with 2D convolutions.

### 2.1 RenderNet

The 3D input encoder consists of an encoder made up of 3D convolutions with channels {8, 16, 16}, kernel sizes {5, 3, 3}, and strides {2, 2, 1} respectively. We add ten 3D residual blocks, before feeding the result of the last block to the projection unit. The unit resizes the tensor from $W\times H\times 32\times 16$ to $W\times H\times (32\cdot 16)$ before feeding it to a $1\times 1$ convolution with the same number of channels. This is followed by ten 2D residual blocks, a $4\times 4$ convolution with $(32\cdot 8)$ channels, and another five 2D residual blocks. To produce the final rendered image, we use a series of 2D convolutions with channels {$32\cdot 4, 32\cdot 2, 32, 16, 3$ (or 1 for greyscale image)}, kernel sizes {4, 4, 4, 4, 4} and strides {1, 2, 2, 2, 1}, respectively. To generate other modalities of the output (for example, in Section 4.1 where we render both the albedo map and normal map), we simply create another branch of 2D convolutions layers starting at the first strided up-convolution layer, and train it jointly with the rest of the network. This allows different modalities to share high-level information, such as object visibility, and only differ in the pixel appearance (shading). This shows the potential to combine training different shading styles into training one model that shares high-level information and separates low-level convolution layers for different shading styles.

### 2.2 Alternative architecture

Here we describe the architecture of the two alternative models EC and EC-deep that are used to compare against RenderNet (see Section 4.2 in the main paper).

The 3D input encoder consists of an encoder made up of 3D convolutions with channels {64, 128, 256, 512}, kernel sizes {4, 4, 4, 4}, and strides {2, 2, 2, 2} respectively. All of these convolutions use

Voxel input   RenderNet   Mesh renderer

Figure 3: Failure cases.

parametric ReLU. This is followed by a fully-connected layer to map the tensors to a 200-dimensional vector and a sigmoid activation function.

For EC, we directly concatenate the lighting and pose parameters to the shape latent vector. For EC-deep, we feed each of them through a fully-connected layer to map each to a 512-dimensional vector. These two vectors are then concatenated to the shape latent vector. The final concatenated vector is then fed through 2 fully-connected layers to map to a 1024-dimensional vector, and another fully-connected layer to map to a $(8 \cdot 8 \cdot 512)$-dimensional vector. The output of this layer is reshaped into a tensor of size $8{\times}8{\times}512$ before being fed to the decoder.

For EC, the decoder consists of 2D convolutions with $4{\times}4$ kernels with channels {512, 512, 256, 256, 128, 128, 64, 64, 32, 32, 16, 1} and strides {2, 1, 2, 1, 2, 1, 2, 1, 2, 1, 2, 1}. For EC-deep, we replace each non-strided convolution in EC with two 2D residual blocks.

### 2.3   Texture decoder

The texture decoder consists of a fully-connected layer to map the 199-dimensional vector input to a vector of size $(32 \cdot 32 \cdot 32 \cdot 4)$, which is then reshaped into a tensor of size $32{\times}32{\times}32{\times}4$. This is followed by a series of 3D convolutions with channels {4, 8, 4}, kernel sizes {4, 4, 4}, and strides {1, 2, 1} respectively. The output is a tensor of size $64{\times}64{\times}64{\times}4$.

## 3   Limitations

RenderNet was trained using mean squared error loss (or binary cross-entropy loss for greyscale images), which tends to create blurry results. The effect is more obvious in certain shaders, such as the Ambient Occlusion. This can potentially be solved by adding an adversarial loss, but we consider this to be future work.

Another potential limitation of our method is the input voxel grid resolution. We mitigate this limitation by training RenderNet on smaller, cropped voxel grids and running inference on larger voxel grids. This is made possible by the fully convolutional design of our architecture. Note that the output size is not limited. In the future, we could leverage data structures such as octrees or different data formats such as unstructured point clouds to further improve the scalability of our model.

As shown in Figure 3, RenderNet has a tendency to over-smoothen sharp diagonal shapes. RenderNet also fails to render extremely thin features, which can easily be handled with the mesh renderer. However, this can be considered to be the limitation of the voxelizing tool, as the input voxel grid fails to capture very thin features, and not of RenderNet itself.