[Reviews · NeurIPS 2018]

Reviewer 1



RenderNet is a neural network that can approximate a 3D render. The network is trained in a supervised way using pairs of inputs (3D model, camera pose and lights) and outputs (3D rendering using an of the shelf standard render). The network has several modules: rigid body transformation (World2camera coordinates), CNN with 3D convolutions, projection Unit (3d to 2D) and finally a CNN for computing the shading. The method is trained in chairs from ShapeNet. However it seams to generalize well for other classes. RenderNet has been used in an application example of inverse graphics (Recovering the 3D structure from a 2D image). There are several works that perform 3D rendering in a differentiable way. The most important ones I knew are already cited in the paper: OpenDR, and the NN based ones. This new method is a more powerful network. It allows 512x512 rendering which is bigger than the previous SOTA. It also allows render texture. And it is differentiable, so it can be used as a module for other networks. As was points, it is difficult to fully understand the paper due to the lack of details. It is not clear to me how the textures are used. I miss details in the method section. Most of the evaluation is qualitative with the figures from the paper. There is only one table with numerical results but it is limited and difficult to understand: What are these methods trained on? What is the testing set? How does it compare to other methods but EC? Why EC-Deep is worse than EC? PSNR for EC and RenderNet are almost the same but visually EC is much worse... Qualitative results of EC-DEEP? What is RenderNet Face? fig 5.c what is that? Regarding the texture it has been only used for faces. Can it generalize to other classes? Is the code going to be available? Either will be very difficult to reproduce.

Reviewer 2



# Summary The paper proposes a novel--fairly straightforward--network architecture to learn image rendering. Several rendering techniques from computer graphics are imitated while still providing a differentiable model. Trained only on chairs, the model is then used to invert the rendering process to reconstruct faces, relight them and show them from a different perspective. # Paper Strengths - Relevant problem: inverse rendering and learning to render tie an important connection between our knowledge of image formation and image semantics. In particular, it allows better reasoning about 3D relations in scenes. - Well written work, clear presentation. - Well chosen experiments to support their idea. - Fairly straightforward idea that smartly picks the fixed and learned parts of the architecture: learning the projection from 3D to 2D allows the model to reason about occlusion. - Although the model is only trained on chairs, it generalizes well to unseen objects. - Ambiguities during the face reconstruction are solved by regularising possible shapes with a pre-trained 3D auto-encoder. # Paper Weakness - The scene representation--a voxel grid or the texture representation--might not fit all use cases and feels limited. # Further Questions - Will the authors release their code upon publication of this paper? - What are the limitations of your work? Voxel volume? Output size? What would make it break? # Conclusion I like the simplicity of the proposed method and the clean presentation of this work. It makes a strong contribution towards inverse rendering.

Reviewer 3



The paper proposes an end-to-end differentiable renderer which takes a voxelized shape as input and outputs the rendered image. Pros: The idea is clean and simple. Compared to previous differentiable renderers, it can condition on lights and textures, and is extensible to inverse rendering tasks. When applying on inverse rendering tasks on human faces, it can generate images of better quality. Cons: 1. The projection unit and 2D convolution (the rendering part) may not be necessary to learn. For example, in Phong shading, the normals can be computed using gradients of the depth map, and the dot operation of the lighting direction and normals can also be differentiable. In Barron & Malik 2015 they already have such differentiable renderer with lighting represented by spherical harmonics. It is also unclear to me why the visibility of objects need to be learned with the projection network. Don't you just need to pick the voxel closest to the camera to know visibility? 2. In Section 4.2, it may not be fair since the other networks need to learn the 3D transformations, while in RenderNet, the 3D transformations are explicitly done. 3. In figure 2, marching cubes is an algorithm to generate a mesh from 3D voxels. I am not sure why it is compared with renderers. 4. It would be more convincing if the reconstructed 3D shape in section 4.3 can be shown in a figure. 5. Lack of quantitative evaluation in section 4.3. I also have some questions about the technical details: 1. I am confused about the output of the RenderNet. In line 136 the paper mentions the output is the shading, but in Figure 1 the output seems to be the Normal Map which is converted to shading. Line 137 also mentions that normal is an optional output. If you directly output shading, do you mean the Phong shading can be learned by the network itself without the need to explicitly specifying S = max(0, l*n + a) as in line 175? In other styles of rendering, do you also directly output the final shading? How do you use the input light position in those cases? Please address the above concerns.